# Reconciling larval and adult sampling methods to model growth across life-stages

**George C. Brooks**[ORCID]*, **Thomas A. Gorman**[¤]**, Yan Jiao, Carola A. Haas**

Department of Fish and Wildlife Conservation, Virginia Tech, Blacksburg, Virginia, United States of America

¤ Current address: Aquatic Resources Division, Washington State Department of Natural Resources, Olympia, Washington, United States of America

* boa10gb@vt.edu

**Data Availability Statement:** All BUGS code and raw data are provided as supplementary files, however owing to the sensitive nature of the species (federally endangered with a high risk of

## Abstract

Individual growth rates are intrinsically related to survival and lifetime reproductive success and hence, are key determinants of population growth. Efforts to quantify age-size relationships are hampered by difficulties in aging individuals in wild populations. In addition, species with complex life-histories often show distinct shifts in growth that cannot be readily accommodated by traditional modelling techniques. Amphibians are often characterized by rapid larval growth, cessation of growth prior to metamorphosis, and resumption of growth in the adult stage. Compounding issues of non-linear growth, amphibian monitoring programs typically sample larval and adult populations using dissimilar methods. Here we present the first multistage growth model that combines disparate data collected across life-history stages. We model the growth of the endangered Reticulated Flatwoods Salamander, *Ambystoma bishopi*, in a Bayesian framework, that accounts for unknown ages, individual heterogeneity, and reconciles dip-net and drift fence sampling designs. Flatwoods salamanders achieve 60% of growth in the first 3 months of life but can survive for up to 13 years as a terrestrial adult. We find evidence for marked variability in growth rate, the timing and age at metamorphosis, and maximum size, within populations. Average size of metamorphs in a given year appeared strongly dependent on hydroperiod, and differed by >10mm across years with successful recruitment. In contrast, variation in the sizes of emerging metamorphs appeared relatively constant across years. An understanding of growth will contribute to the development of population viability analyses for flatwoods salamanders, will guide management actions, and will ultimately aid the recovery of the species. Our model formulation has broad applicability to amphibians, and likely any stage-structured organism in which homogenous data cannot be collected across life-stages. The tendency to ignore stage-structure or omit non-conforming data in growth analyses can no longer be afforded given the high stakes of management decisions, particularly for endangered or at-risk populations.

## Introduction

Individual growth rates are intrinsically related to survival and lifetime reproductive success and are key determinants of population growth [1–3]. Understanding an organism's pattern of

illegal collection for the pet trade), GPS points of site locations cannot be revealed.

**Funding:** This work was supported by the USDA National Institute of Food and Agriculture, McIntire Stennis project 1006328, awarded to CAH. The funders had no role in study design, data collection and analysis, decision to publish, or preparation of the manuscript.

**Competing interests:** The authors have declared that no competing interests exist.

growth through time is of paramount importance to evolutionary and ecological studies and is necessary to construct realistic population dynamic models [4–6]. Without basic demographic and life-history data, projection models and viability analyses used in conservation management are marred with uncertainty [7, 8]. Furthermore, imprecise growth estimates are likely to bias rates of population growth and extinction probabilities [5, 9].

If animals can be accurately aged, there are many choices for those wishing to model growth, however difficulty in aging individuals from wild populations is a salient problem in ecological studies [10–14]. Techniques that use physical characteristics (skeletochronology, otoliths, etc.) are often lethal, and size-frequency data are highly unreliable, particularly in species with overlapping generations [15]. As a result, several models have been modified to estimate growth rates for individuals of unknown age from mark-recapture data [16–17]. In the context of endangered species management, mark-recapture studies may offer the only practical method to accurately model growth.

Of the models that have been adapted to mark-recapture data, the von Bertalanffy growth equation (VBGE) has yielded the most applications and model developments aimed at achieving increased biological realism. The VBGE benefits from relatively few model parameters to estimate and can be derived from metabolic theory [18, 19]. The VBGE has been modified to include seasonal or inter-annual fluctuations in growth rates [20, 21], individual heterogeneity within populations [22–24], and shifts in growth as a result of stage-structured life histories [25].

To date, modelling shifts in growth requires data collected across life-stages to be homogenous. Whilst this assumption may hold for data collected to detect seasonal trends in growth, this is altogether less common in studies of organisms that exhibit distinct life history stages. In many taxa, life stages are distinct enough to warrant different sampling techniques, and data are often collected piecemeal across an organism's ontogeny [13, 15]. Pond breeding amphibians are a case in point. Sampling methodologies for aquatic larvae are necessarily different from those employed to monitor adults [26–28]. Moreover, many larval amphibians are small enough to preclude unique marking methodologies, or marks may be lost as a result of metamorphosis [29]. Hence, data collected for larvae and adults are sometimes not only qualitatively, but quantitatively different. Although the two types of data are disparate, they are not independent, and thus growth models can be strengthened by use of a single framework.

Here we model the growth of the federally endangered Reticulated Flatwoods Salamander (hereafter flatwoods salamanders), *Ambystoma bishopi*, using a hierarchical Bayesian approach. Flatwoods salamanders inhabit longleaf pine flatwoods in the southeastern Coastal Plain in the United States. Adults are fossorial and occupy mesic upland habitats, and undertake annual migrations to ephemeral wetlands with well-developed herbaceous groundcover to breed [30–33]. Flatwoods salamanders lay their eggs in dry wetland basins before they fill, allowing embryos to develop so eggs can hatch when wetlands are inundated [32, 34, 35]. Unpredictable precipitation regimes and pond-filling dates favor such a delayed development strategy [36, 37].

Here we employ a modified von Bertalanffy equation with additional latent parameters for age and size at metamorphosis to 1) estimate growth rates in flatwoods salamanders, 2) model shifts in growth across the metamorphic transition, and 3) reconcile disparate data types collected from different sampling methodologies into a unified modelling framework. The final model incorporates individual heterogeneity in adult growth and variability in the timing and size at metamorphosis. Typical of many salamanders, the majority of flatwoods salamander growth occurs in the larval stage, but the majority of an individual's lifetime is spent as a terrestrial adult. We find evidence for marked variability in growth rate, the timing and age at metamorphosis, and maximum size, within populations. An understanding of growth will

contribute to the development of population viability analyses for flatwoods salamanders, will guide management actions, and will ultimately aid the recovery of the species.

## Materials and methods

Measurements were obtained from a long-term mark recapture study of two breeding populations of flatwoods salamanders on Eglin Air Force Base, Florida. Wetlands were completely encircled with drift fences constructed from 60 cm tall metal flashing buried in the sediment approximately 15–20 cm. 85 cm x 20 cm funnel traps were placed flush with the fence and ground at approximately 10 m intervals on both sides of the fence. Drift fences were run discontinuously from 2010 to 2019. Fence operations began each fall to coincide with rains that trigger breeding migrations in late October or early November. Once we initiated drift fence operations, we checked traps multiple times per night through March-May, depending on how long wetlands continued to hold water. Upon capture, we recorded the date and time of capture, and measured snout vent length (SVL) to the nearest millimeter. All animals were marked with passive integrated transponder (PIT) tags. Following processing, animals were released on the opposite side of the fence to which they were caught.

In addition to measurements of terrestrial adults, measurements of larval salamanders were obtained through long-term dipnet sampling at all known breeding wetlands across Eglin AFB (n = 13), collected from 2003 to 2019. All measurements of adult and larval animals were taken using calipers. Sites were sampled using Model SH-2 and SH-2D (Mid-Lakes Corporation, Knoxville, TN) dipnets and efforts were concentrated in areas with inundated herbaceous vegetation [38]. All research was approved by the Virginia Tech Institutional Animal Care and Use Committee, protocol 19–113.

We employed a Bayesian hierarchical model to investigate individual growth in flatwoods salamanders. The Bayesian framework accommodates multiple sources of uncertainty and can include individual heterogeneity in growth rates [21, 39, 40]. When growth parameters vary within a population, hierarchical models allow for the estimation of individual-specific growth trajectories whilst still drawing on information from the population as a whole for statistical power [13]. In addition, the hierarchical approach lends itself to inconsistent capture histories prevalent in mark-recapture studies, and can incorporate as much or as little prior knowledge of the organism's biology as deemed appropriate.

### Model for larval stage growth

For larval measurements obtained via dipnet surveys, we employ the traditional formulation of the von Bertalanffy equation, using date of pond-filling as an estimate of hatch date (age = 0) for each individual. The von Bertalanffy equation is most commonly applied in studies of ectothermic vertebrates, but is considered a universal model of growth, and strongly resembles curves derived from basic metabolic principles [19]. For a larval individual of age $t$, the predicted size $L_t$ from the von Bertalanffy equation is expressed as:

$$L_t = L_\infty (1 - e^{-k^L(t-t_0)})$$

$$L_\infty \sim U(30, \ 100)$$

$$t_0 \sim U(-2, \ 2)$$

$$k^L \sim G(0.1, \ 0.1)$$

where $k^L$ represents the growth-rate parameter of larval stage individuals, $L_\infty$ the asymptotic size in mm (across all parameterizations), and $t_0$ the theoretical age in years when length = 0. $L_\infty$ was assigned a uniform prior with vague bounds based on the maximum and minimum sizes recorded for the species, $k_L$ was assigned a vague Gamma prior, and $t_0$ a uniform prior with vague bounds.

## Metamorphic transition

To integrate the larval and adult sub-models, measurements from 766 metamorphs were incorporated into the analysis to estimate size and the corresponding age at metamorphosis. Using parameters estimated from the larval model, predicted size at metamorphosis is thus defined:

$$L_t = L_\infty(1 - e^{-k^L(t_m - t_0)})$$

$$t_m \sim U(0.2, 0.7)$$

where $t_m$ is age at metamorphosis. $t_m$ is assigned a uniform prior with bounds based on published data on larval flatwoods salamander developmental rates [33].

## Model for adult stage growth

For terrestrial adults repeatedly sampled at drift fences, we use a modified version of the von Bertalanffy equation that can accommodate mark-recapture information from individuals of unknown age. For the initial capture occasion, length is modelled similarly to larvae, but with the initial size set to the predicted length at metamorphosis,

$$L_t = L_\infty - (L_\infty - L_{t_m})(e^{-k^A(t - (t_m + t_0))})$$

$$k^A \sim G(0.1, 0.1)$$

$$t \sim logN(log(\alpha), \sigma_t^2)$$

$$\alpha \sim U(0.5, 20)$$

$$\sigma_t^2 \sim G(0.1, 0.1)$$

As age of adult individuals at first capture are unknown, they must be estimated and are assumed to be drawn from a truncated lognormal distribution. $L_\infty$ was assigned a uniform prior with vague bounds based on the maximum and minimum sizes recorded for the species, and $k^A$ was assigned a vague Gamma prior. For all subsequent occasions, length-at-age relationships are modelled using the difference in time between capture occasions, $\delta t$, such that:

$$L_t = L_\infty - (L_\infty - L_{t_m})(e^{-k^A((t + \delta t) - (t_m + t_0))})$$

By estimating the unknown age of adults and including parameters for size/age at metamorphosis, the larval, metamorph, and adult data can be reconciled into a single modelling framework and sampled jointly from the posterior. Following Hatch and Jiao (2016), observed lengths ($L_{obs}$) are assumed to be drawn from a normal distribution with mean $L_t$ and variance $\sigma_L^2$, to account for measurement error and/or individual heterogeneity in growth rates. $\sigma_L^2$ is

assigned a vague gamma prior, and is used across all life-stages in the model:

$$L_{obs}|L_t,\ \sigma_L^2\ \sim\ N(L_t,\sigma_L^2)$$

$$\sigma_L^2\ \sim\ G(0.1,\ 0.1)$$

All models were fitted in R and WinBUGS using Markov chain Monte Carlo (MCMC) optimization [41–43]. Three chains of MCMC samples were generated from the posterior distributions of the model parameters, each of length 500,000 with the first 100,000 values being discarded as burn-in. To minimize autocorrelation, only every 100th sample was drawn for posterior summaries. Adequate convergence was assessed using Gelman-Ruben diagnostics and inspection of trace plots [44]. Bayesian p-values were calculated to assess goodness-of-fit [44]. All reported point estimates are posterior means, with associate 95% credible intervals in parentheses.

## Results

Through dipnet sampling, 411 larval measurements were obtained from 2010 to 2018 between the months of December and April (Fig 1). Larval sizes ranged from 3.9 mm to 42.7 mm. Through drift fence sampling, 766 metamorphs and 927 adult salamanders were captured and marked. Metamorph size ranged from 27.5 mm to 51.9 mm, and showed considerable within- and between-year variability (Fig 2). Of the adults, 373 were recaptured on at least one occasion, and SVL ranged from 37.5–78.2mm, averaging 58.4mm (Fig 3).

All model parameters adequately converged; all potential scale reduction factors (PSRF) for individual parameters were < 1.1. The multivariate PSRF for the full model was 1.08. Posterior p-values for larval size and adult growth increments both approximated 0.5 (0.51 and 0.506 respectively), indicating a good model fit. All parameters achieved an effective sample size >500 in the MCMC chains. Parameter estimates were insensitive to prior specification.

Larvae grew rapidly ($k^L$ = 1.77; CI: 1.65–1.91), reaching sizes necessary for metamorphosis (~35mm SVL) within 18 weeks (CI: 15–20; Figs 1 and 4). Timing and size at metamorphosis were positively correlated (Fig 2) and exhibit marked variability across cohorts. Transition between the two life stages occurred when individuals averaged 39.3 mm (CI: 37.8–40.1; Figs 2 and 4). Following metamorphosis, adults grew at an initial growth rate ($k^A$) of 0.91 (CI: 0.73–1.13), corresponding to approximately 6 mm of growth in the first year. Adults grew to an average of 59.0 mm SVL (CI: 57.9–60.0), however there was significant variation among individuals in asymptotic size ($\sigma_{L_\infty}^2\ =\ 5.0$; Fig 4).

Individual heterogeneity and unknown ages of adult individuals contributed the greatest sources of uncertainty. Magnitudes of error in observed lengths sometimes exceeded 5% of the actual measurement, making it difficult to partition out model uncertainty from true variation among individuals. Maximum longevity for the species was estimated to be 13 (Fig 5), however growth plateaued at approximately 7 years of age, and thus uncertainty in age estimates for older/fully grown individuals was high. Credible intervals for the theoretical age at size = 0 ($t_0$) overlapped with zero, and thus do not influence interpretation of model parameters.

## Discussion

Here we present the first multistage growth model that combines disparate data collected across life-history stages. Measurements from larval surveys and mark-recapture data from drift-fence studies were reconciled into a single modeling framework. Estimating growth from only older individuals or failing to accurately quantify uncertainty can severely bias estimates

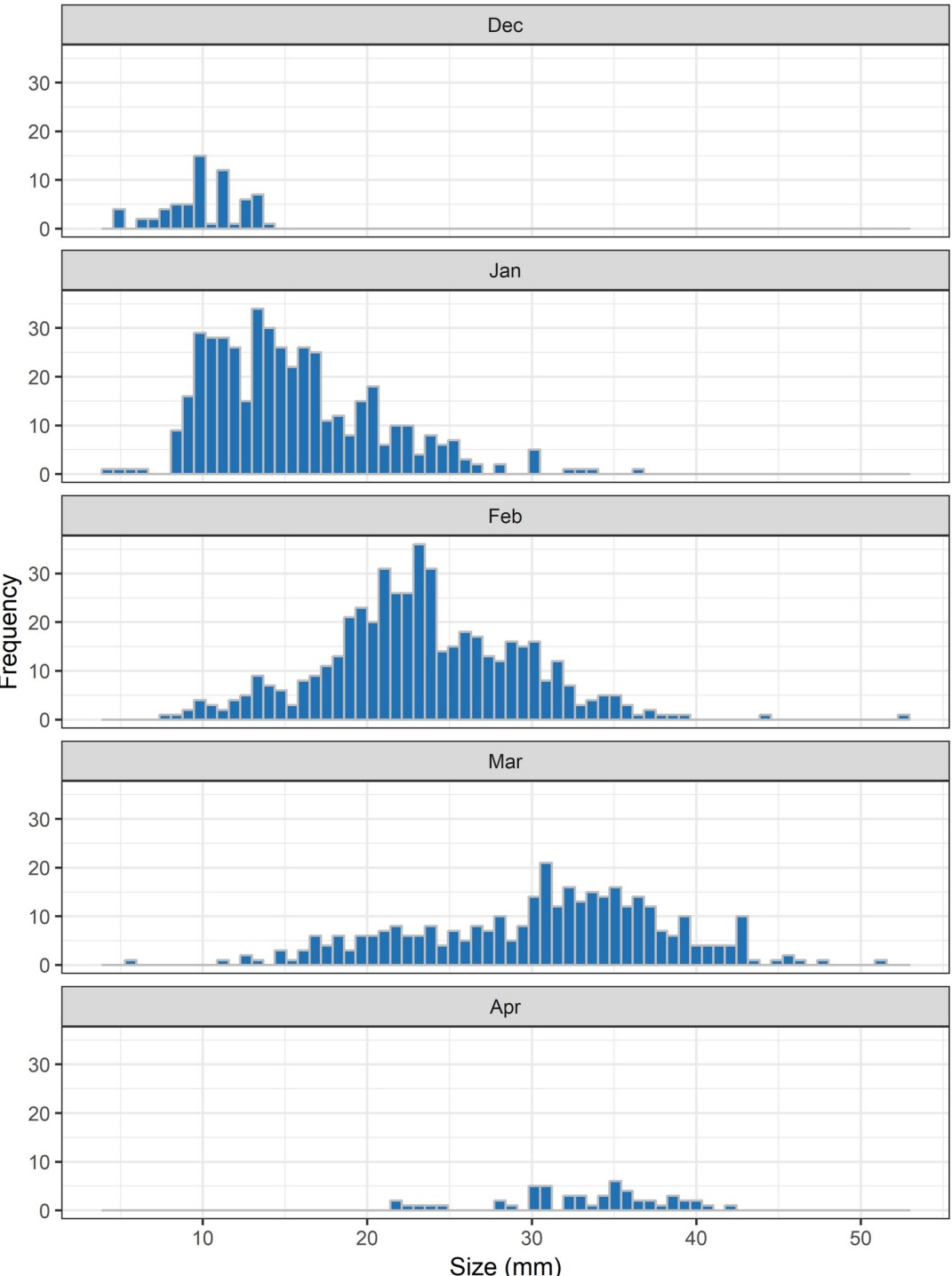

**Fig 1. Larval size distribution.** Size distribution of reticulated flatwoods salamander larvae captured by dipnetting and spotlighting at approximately 8–12 wetlands, including the two drift-fenced wetlands, on Eglin Air Force Base, Florida. Measurements are pooled across the years 2010–2018.

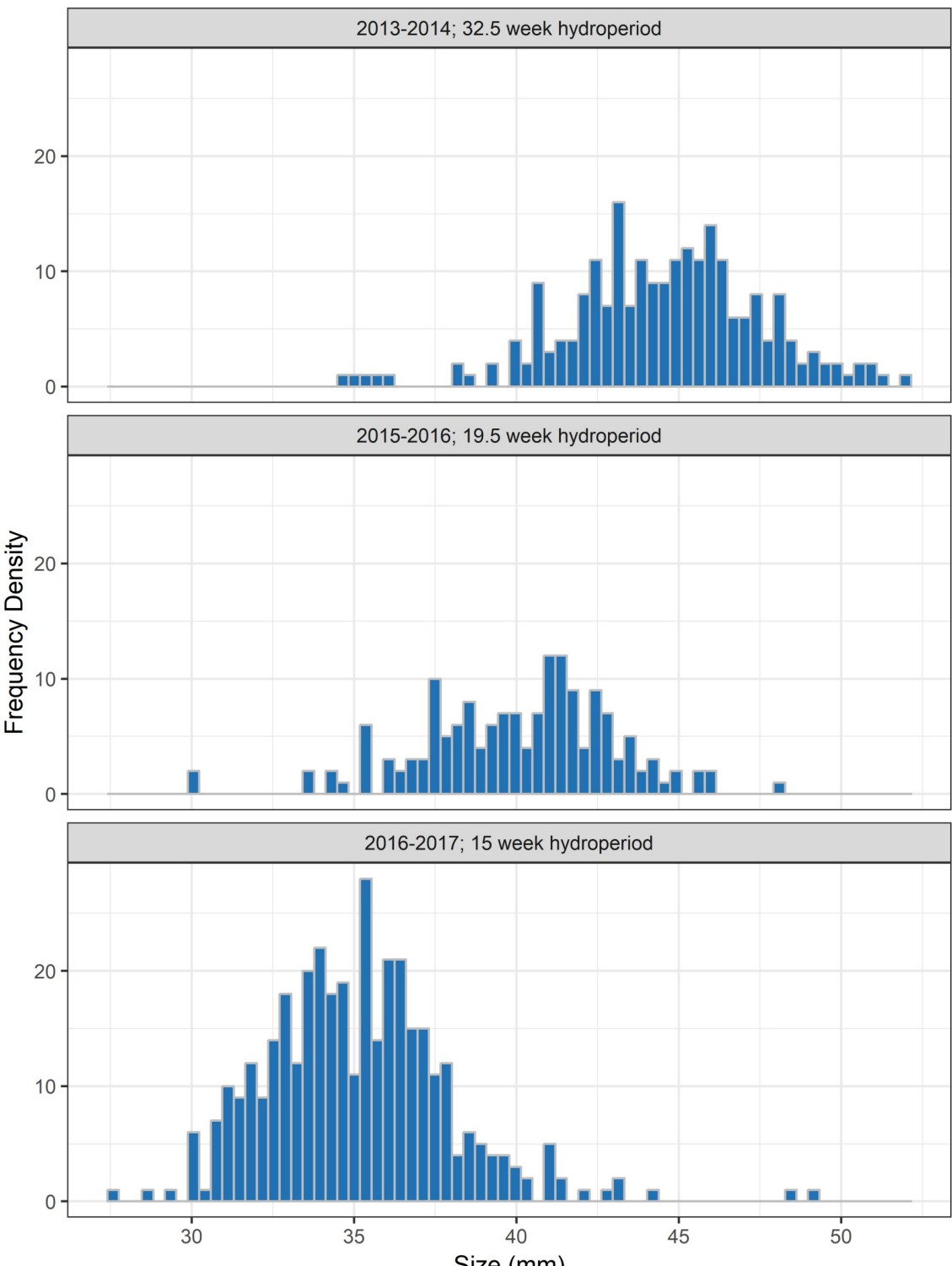

**Fig 2. Metamorph size distribution.** Size distribution of reticulated flatwoods salamander metamorphs captured emigrating from two drift-fenced wetlands on Eglin Air Force Base, Florida, for a subset of years with differing hydroperiods. The year and corresponding hydroperiod for the wetlands that produced these cohorts are displayed above each panel.

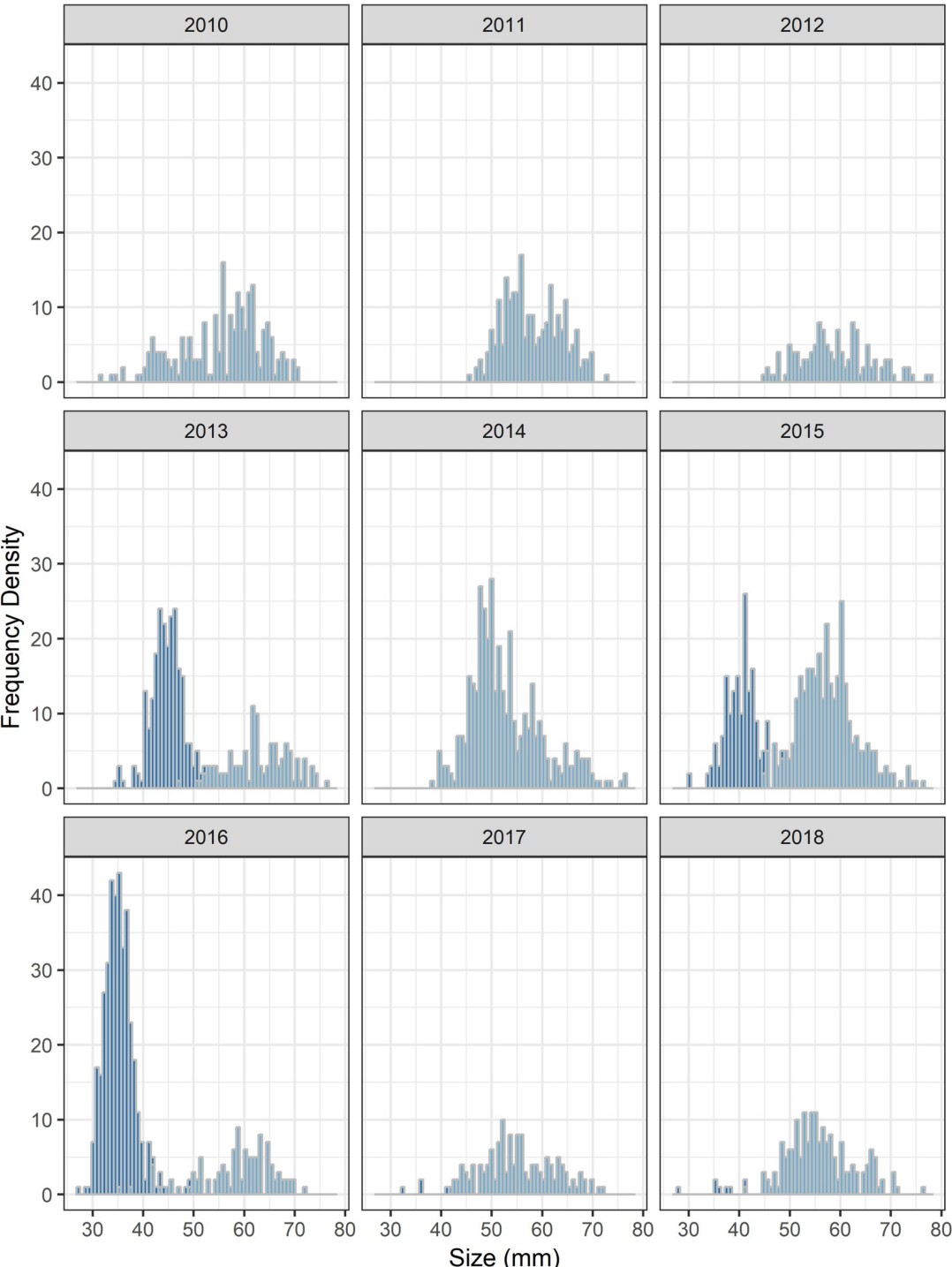

**Fig 3. Adult size distribution.** Size distribution of reticulated flatwoods salamanders captured as terrestrial forms at two drift-fenced wetlands on Eglin Air Force Base, Florida, across years. Histogram of snout-vent length (SVL) for post-metamorphic individuals by breeding season. Data are partitioned into adults (light blue bars), and yearlings (dark blue bars).

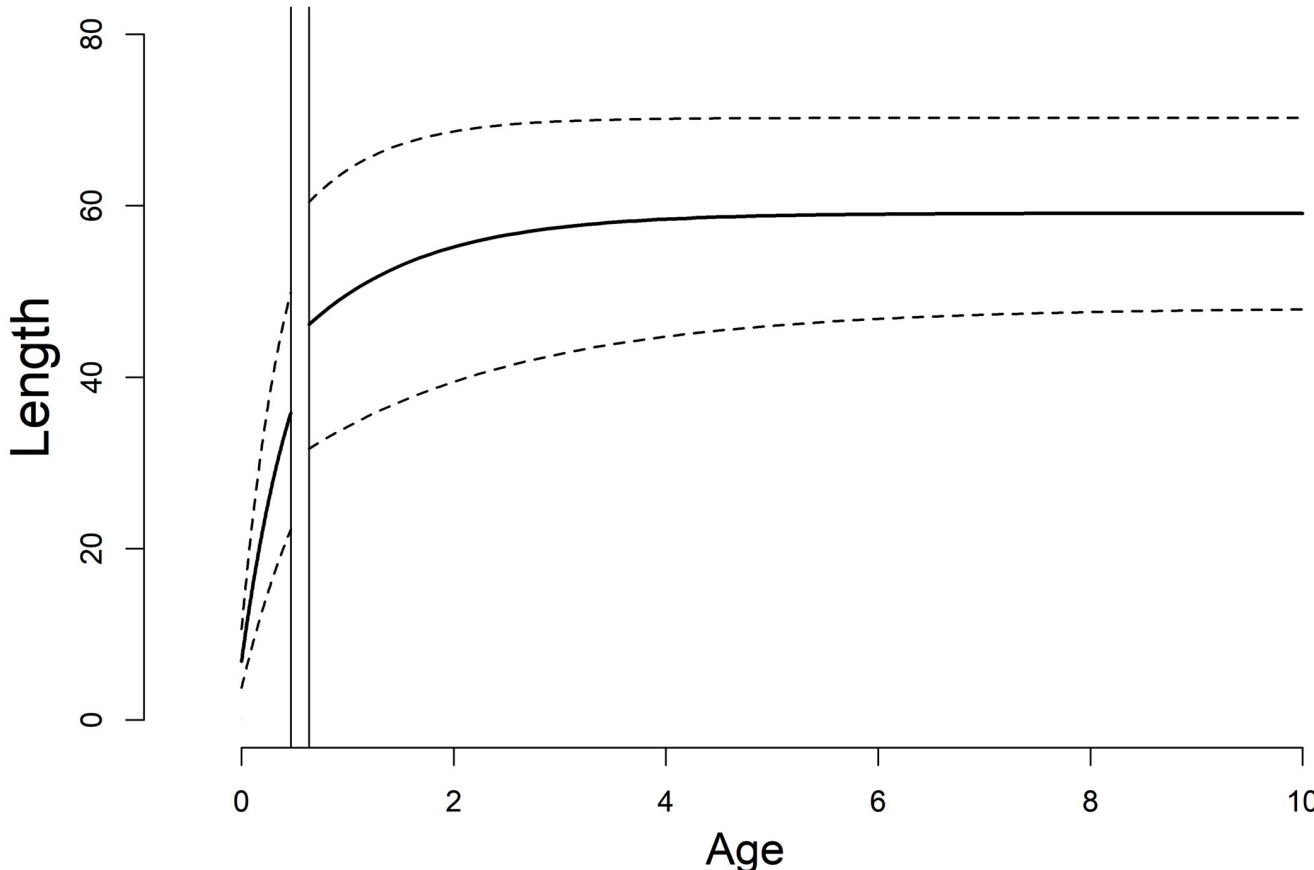

**Fig 4. Growth curve.** Predicted growth of reticulated flatwoods salamanders across both larval and adult stages. The two vertical lines represent the 95% credible intervals for minimum and maximum age at metamorphosis respectively, and the dashed lines represent 95% posterior predictive intervals. Length is snout-vent length (SVL) in mm.

[10, 14, 45]; the Bayesian approach presented here, permits the inclusion of all available data across life-stages whilst accounting for multiple sources of uncertainty in parameters of interest.

Like other ambystomatids, the majority of growth in flatwoods salamanders occurs in the larval stage; individuals reach ~60% of their asymptotic size prior to metamorphosis. Larval growth is an order of magnitude faster than that of adults. Variability in larval growth likely results in only some individuals within a cohort attaining sizes necessary for metamorphosis prior to pond-drying [46, 47]. Half of all larval growth trajectories are not steep enough to reach sizes (>35mm) necessary to successfully transition across life-stages (Fig 1). As we did not have repeated measures from dipnet sampling however, we were unable to include individual heterogeneity directly into the larval model parameterization.

Variation in larval growth within a cohort may pale in comparison to variation in size at metamorphosis across years (Fig 2). The distribution of sizes at metamorphosis appears to correspond to the hydroperiod duration in a given year. If wetlands continue to hold water, it appears as in other species (e.g. [48]), larvae will postpone metamorphosis in favor of continued growth. This plasticity in timing holds important consequences for population viability, as increased size at metamorphosis is thought to confer fitness benefits for the remainder of an individual's lifetime [46, 48–51] (but see [52]). Knowledge of this innate plasticity will prove

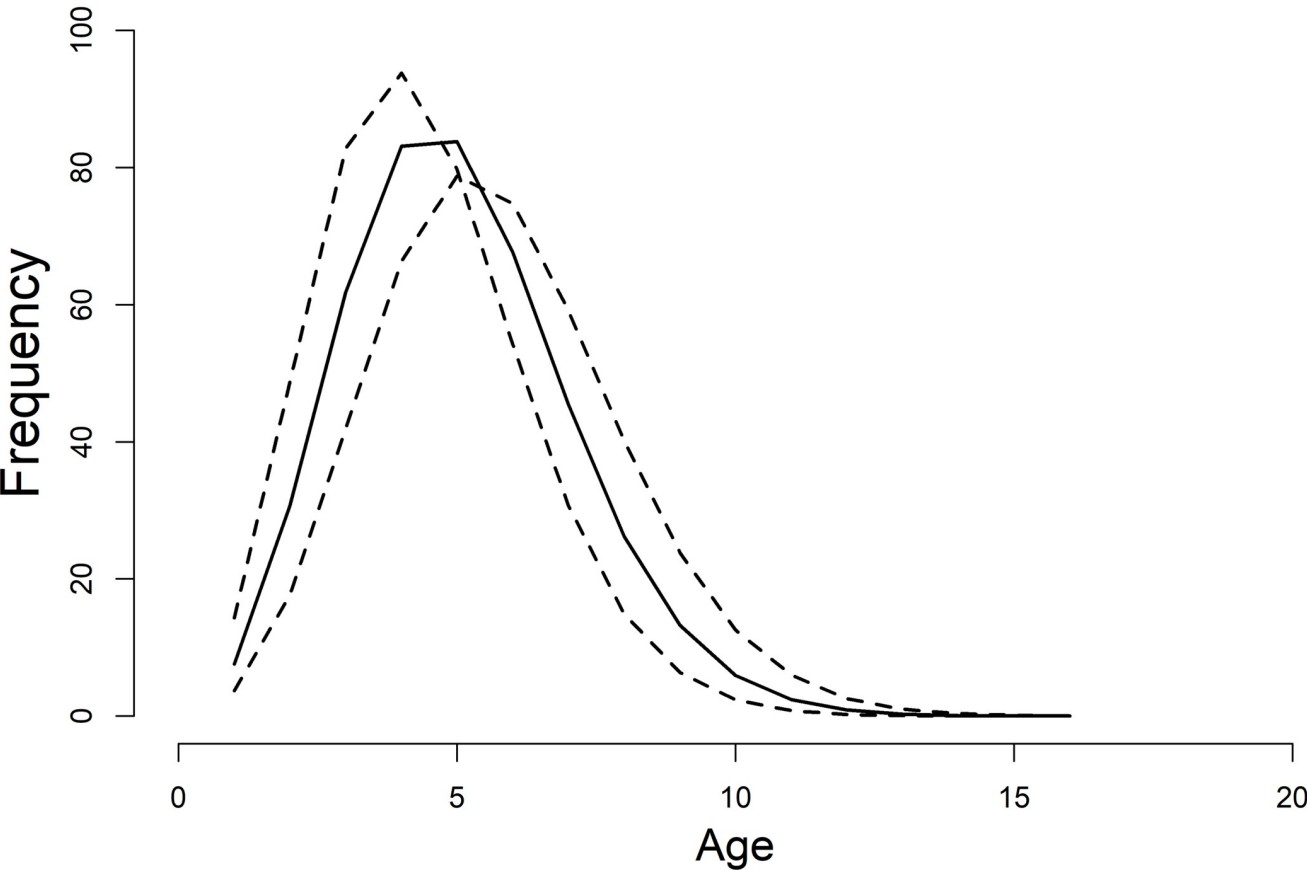

**Fig 5. Predicted age distribution.** Predicted age distribution of adult flatwoods salamander at the start of the study. Average age of the population was between 4 and 5 years old; maximum longevity is 12–13 years. Dashed lines reflect 95% credible intervals.

invaluable to captive rearing efforts, and presents a facet of the life-history that can be targeted by management, through artificial manipulation of pond hydroperiods [53, 54].

Following metamorphosis, growth of terrestrial adults slowed considerably and plateaued after approximately 7 years at 60mm, but was highly variable among individuals. As we were unable to track individuals across the metamorphic transition, it remains to be seen as to whether variability in adult growth stems from variability in the larval stage, but concomitant individual rates across stages represents the most parsimonious explanation. Flatwoods salamanders are comparatively small for the family Ambystomatidae [55], and this may simply reflect a shorter larval development time, resulting in smaller individuals at metamorphosis. Metamorphs of congeners regularly exceed 50mm in length [55], in contrast to the focal species that rarely grew to 45mm prior to metamorphosis.

Our results also suggest flatwoods salamanders are not as long lived as other Ambystomatids. No individual was determined to be more than 12 years old. For any species that grows asymptotically, estimating ages for individuals that are at or close to maximum size is challenging, and thus wide confidence intervals on predicted ages of larger individuals prevents any strong conclusions. Survival rates in flatwoods salamanders however, do appear lower than congeners (unpublished data, G. Brooks). It is unclear whether lower longevity reflects a naturally faster life history strategy in flatwoods salamanders compared to closely related species, or a sign of inflated mortality that contributes to the imperiled status of the species. Diagnosing

the primary agents of mortality in terrestrial individuals is of paramount importance, as it may reveal the cause of declines, and in turn hold the key for species recovery.

Discerning growth rates and size-age relationships can facilitate conservation efforts for threatened and endangered species [17, 56]. From such metrics, one can derive stable age distribution, age at maturity, and longevity, all of which strongly influence estimates of population growth rates from viability analyses [57]. Stage-structured organisms pose a real challenge to this end, and as a result, previous studies are largely limited to species without distinct life-stages [22, 58–61], or for which homogenous data across stages can be collected [25, 62]. For all other circumstances, the tendency is to ignore stage-structure or to omit non-conforming data. We argue however, that for rare taxa, researchers cannot afford these concessions, as even data collected piecemeal contains real ontological insight and utility. Reliable population projections require accurate measures of growth rates across all life stages. Our model formulation has broad applicability to amphibian studies and studies of other stage-structured organisms in which homogenous data cannot be collected across life-stages. Given the ubiquity of complex life-histories and the logistical constraints of monitoring organisms throughout ontogeny, our approach will prove useful for a variety of ecological studies, extending far beyond amphibians. For flatwoods salamanders specifically, an understanding of growth will contribute to the development of population viability analyses, will improve management decisions and actions, and will aid the recovery of the species.

## Supporting information

**S1 Appendix. Raw data and BUGS code.**
(ZIP)

## Acknowledgments

We would like to thank the team of people who have made this research possible. Special mention should be given to Kelly Jones, Brandon Rincon, Steve Goodman, Vivian Porter, and the myriad seasonal technicians involved in data collection. We thank Emmanuel Frimpong for his guidance in developing the statistical analyses. We thank the Natural Resources Branch of Eglin Air Force Base (Jackson Guard), the U.S. Fish and Wildlife Service, the Department of Defense Legacy Resource Management Program, the Florida Fish and Wildlife Conservation Commission, the Aquatic Habitat Restoration and Enhancement Program, and the Department of Fish and Wildlife Conservation at Virginia Tech for funding and logistical support on this project.

## Author Contributions

**Conceptualization:** George C. Brooks, Thomas A. Gorman, Carola A. Haas.

**Data curation:** George C. Brooks.

**Formal analysis:** George C. Brooks, Yan Jiao.

**Funding acquisition:** Thomas A. Gorman, Carola A. Haas.

**Investigation:** George C. Brooks, Thomas A. Gorman, Carola A. Haas.

**Methodology:** George C. Brooks, Thomas A. Gorman, Yan Jiao.

**Project administration:** Carola A. Haas.

**Resources:** Carola A. Haas.

**Supervision:** Thomas A. Gorman, Yan Jiao, Carola A. Haas.

**Writing – original draft:** George C. Brooks.

**Writing – review & editing:** George C. Brooks, Thomas A. Gorman, Yan Jiao, Carola A. Haas.

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
