## [Decision Letter · Decision Letter 0]

5 Jun 2020

PONE-D-20-12247

Reconciling larval and adult sampling methods to model growth across life-stages

PLOS ONE

Dear Dr. Brooks,

Thank you for submitting your manuscript to PLOS ONE. After careful consideration, we feel that it has merit but does not fully meet PLOS ONE’s publication criteria as it currently stands. Therefore, we invite you to submit a revised version of the manuscript that addresses the points raised during the review process.

Two external referees and me have now evaluated your manuscript. Both reviewers provide copious suggestions for improvement, including providing further clarification of methods and sampling techniques, improving the organization of the presentation, and providing the BUGS code described in the MS. I think the MS provides some potentially important methods and their implementation, but the MS can be substantially improved prior to possible publication.

We look forward to receiving your revised manuscript.

Kind regards,

William J. Etges

Academic Editor

PLOS ONE

Journal Requirements:

2. In your Methods section, please provide additional location information of the collection sites, including geographic coordinates for the data set if available.

3. Thank you for including your ethics statement: 

"This research was approved by IACUC, protocol 19-113.".   

a. Please amend your current ethics statement to include the full name of the ethics committee that approved your specific study.

For additional information about PLOS ONE submissions requirements for ethics oversight of animal work, please refer to http://journals.plos.org/plosone/s/submission-guidelines#loc-animal-research  

Reviewers' comments:

Reviewer's Responses to Questions

**Comments to the Author**

1. Is the manuscript technically sound, and do the data support the conclusions?

Reviewer #1: Yes

Reviewer #2: Partly

2. Has the statistical analysis been performed appropriately and rigorously? 

Reviewer #1: I Don't Know

Reviewer #2: Yes

3. Have the authors made all data underlying the findings in their manuscript fully available?

Reviewer #1: Yes

Reviewer #2: No

4. Is the manuscript presented in an intelligible fashion and written in standard English?

Reviewer #1: Yes

Reviewer #2: Yes

5. Review Comments to the Author

Reviewer #1: Overall, I thought this was an interesting manuscript that uses contemporary modeling methods to better understand growth and age in an imperiled vertebrate species. Authors provide a novel modeling framework that holds great utility, especially for researchers that study animals with complex life cycles. The research also has the potential to aid in endangered species recovery through a better understanding of life history. However, I have several comments below that I believe will improve the manuscript. Specifically, authors often rely, too heavily, on citations to relate their findings to previous work. There is nothing wrong with being explicit! R/WinBUGS Code should be provided as supplemental material (and referenced in the methods section). I am not an expert on growth models presented here, and although I have confidence in the authors’ abilities, I cannot comment on the model in great detail.

Other comments

Abstract: Provide some results! Currently, results of quite general (i.e., marked variability in growth, the timing and age at metamorphosis, etc.)

Line 27: Remove “hence”

Line 34: Topic sentence and supported sentences are mis-matched. The topic sentence indicates the paragraph will focus on the “many choices for those wishing to model population growth.” The supporting sentences do not focus on these model choices. Rather the focus is on “difficulty aging” and “techniques.”

Line 40-41: Why are mark-recapture studies especially important for endangered species management? Is it related to non-lethal nature of capture-mark-recapture studies?

Line 49: Remove “however”

Line 87: I don’t recall reading details on drift fence design or protocols used. Did the driftnet completely encircle each pond(s)? When and how often were they checked? Spacing between buckets? How were animals marked? Are you using Total length or Snout-to-vent length?

Lines 92-97: I would think most readers understand the benefits of using a Bayesian hierarchical model. These benefits are sufficiently outlined by the authors. What isn’t clear, however, is if the authors are basing their model on the Eaton and Link model (i.e., citation 13). If so, an explanation of the Eaton and Link model is needed as well as any modification to this model. Additional information, such as the importance of accounting for measurement error, should also be highlighted.

Lines 101-157: I’m not an expert on growth models. However, I think the authors provide a decent explanation of their model. Choice of priors was not explained and may be needed?

Lines 163-169: This paragraph should be the first one in the Results section as it explains the raw data as opposed to the modeling.

Lines 185: What sizes are needed to reach metamorphosis? Can you just provide the size?

Line 188-189: “Some evidence that size of metamorphosis impacted return rates the following year (Fig 3), possibly of reduced survival of smaller metamorphs”. First, I have difficulty interpreting this from Figure 3. Second, seems like this speculation (and lines 190-191) is best left for the discussion, if at all.

Line 192: “…from larvae to adult..” Should this be larvae to metamorph?

Lines 206-207: “Similarly, discernment of the demographic structure…” Again, should this be in the discussion section?

Lines 229-230: What are the sizes necessary to successfully transition across life-stages?

Line 241 – remove “this” as it’s written twice

Line 255 – What are some of the ages obtained by other Ambystomatids? And survival estimates.

Line 261 – The citation [51] is from Stearns, but it is being used as a comparison to related another species to flatwoods salamanders. This is not useful. Please be explicit in this comparison.

Line 265 – It is not clear if this topic sentence is based on your results? Who are the “individuals” you are discussing. Do the authors provide results that suggest distinct seasonal patterns? Where?

Line 291 – remove “however”

Reviewer #2: In general, this manuscript was technically sound, but important details regarding field and some analytical methods were missing. The data were not available, which is understandable for such a rare species, but some other information, particularly the BUGS model code, would be very useful to readers. The manuscript was generally well-written, but I have provided a few comments regarding organization and grammar in the attached document.

Please see attached file for my full review.

6. PLOS authors have the option to publish the peer review history of their article (what does this mean?). If published, this will include your full peer review and any attached files.

Reviewer #1: No

Reviewer #2: No

---

## [Author Response · Author response to Decision Letter 0]

13 Jun 2020

We would like to start by thanking the reviewers for their comments. I think they have greatly improved the manuscript. Following suggestions, we have provided more details of methods and sampling techniques, addressed reviewers’ comments to improve clarity and organization, and providing the BUGS code and raw data. Responses to specific comments follow.

AE comments:

We believe we have corrected any and all formatting errors 

2. In your Methods section, please provide additional location information of the collection sites, including geographic coordinates for the data set if available.

We have added additional information about the study location, although specific GPS points cannot be revealed due to security concerns. (The study species is federally endangered and threatened with illegal collection for the pet trade.)

3a. Please amend your current ethics statement to include the full name of the ethics committee that approved your specific study.

corrected

3b. Once you have amended this/these statement(s) in the Methods section of the manuscript, please add the same text to the “Ethics Statement” field of the submission form

done

We have included the raw data and BUGS code as a supplementary file

Reviewer #1: 

Overall, I thought this was an interesting manuscript that uses contemporary modeling methods to better understand growth and age in an imperiled vertebrate species. Authors provide a novel modeling framework that holds great utility, especially for researchers that study animals with complex life cycles. The research also has the potential to aid in endangered species recovery through a better understanding of life history. 

Thanks! We think it will be useful too!

Abstract: Provide some results! Currently, results of quite general (i.e., marked variability in growth, the timing and age at metamorphosis, etc.)

We have added some specific results to the abstract

Line 27: Remove “hence”

done

Line 34: Topic sentence and supported sentences are mis-matched. The topic sentence indicates the paragraph will focus on the “many choices for those wishing to model population growth.” The supporting sentences do not focus on these model choices. Rather the focus is on “difficulty aging” and “techniques.”

we have revised the opening sentence to improve clarity

Line 40-41: Why are mark-recapture studies especially important for endangered species management? Is it related to non-lethal nature of capture-mark-recapture studies?

yes, as stated in the previous sentence of that paragraph

Line 49: Remove “however”

done

Line 87: I don’t recall reading details on drift fence design or protocols used. Did the driftnet completely encircle each pond(s)? When and how often were they checked? Spacing between buckets? How were animals marked? Are you using Total length or Snout-to-vent length? 

we have added a description of the long-term drift fence study and methodologies.

Lines 92-97: I would think most readers understand the benefits of using a Bayesian hierarchical model. These benefits are sufficiently outlined by the authors. What isn’t clear, however, is if the authors are basing their model on the Eaton and Link model (i.e., citation 13). If so, an explanation of the Eaton and Link model is needed as well as any modification to this model. Additional information, such as the importance of accounting for measurement error, should also be highlighted.

The model is not based on Eaton and Link, so no description necessary, but a sentence highlighting the importance of measurement error has been added

Lines 101-157: I’m not an expert on growth models. However, I think the authors provide a decent explanation of their model. Choice of priors was not explained and may be needed?

We have added a sentence in the results addressing/explaining choice of priors 

Lines 163-169: This paragraph should be the first one in the Results section as it explains the raw data as opposed to the modeling.

done

Lines 185: What sizes are needed to reach metamorphosis? Can you just provide the size?

included

Line 188-189: “Some evidence that size of metamorphosis impacted return rates the following year (Fig 3), possibly of reduced survival of smaller metamorphs”. First, I have difficulty interpreting this from Figure 3. Second, seems like this speculation (and lines 190-191) is best left for the discussion, if at all.

removed

Line 192: “…from larvae to adult..” Should this be larvae to metamorph?

corrected

Lines 206-207: “Similarly, discernment of the demographic structure…” Again, should this be in the discussion section?

removed

Lines 229-230: What are the sizes necessary to successfully transition across life-stages?

added clarification

Line 241 – remove “this” as it’s written twice

corrected

Line 255 – What are some of the ages obtained by other Ambystomatids? And survival estimates.

this is addressed in subsequent paragraphs in the discussion

Line 261 – The citation [51] is from Stearns, but it is being used as a comparison to related another species to flatwoods salamanders. This is not useful. Please be explicit in this comparison.

Removed erroneous citation

Line 265 – It is not clear if this topic sentence is based on your results? Who are the “individuals” you are discussing. Do the authors provide results that suggest distinct seasonal patterns? Where?

whole paragraph has been removed following reviewer 2 comment

Line 291 – remove “however”

whole paragraph has been removed following reviewer 2 comment

 

Reviewer #2: 

In general, this manuscript was technically sound, but important details regarding field and some analytical methods were missing. The data were not available, which is understandable for such a rare species, but some other information, particularly the BUGS model code, would be very useful to readers. 

Thanks! Yes, we definitely cannot release detailed site information owing to risk of illegal collection for the pet trade, but we can detail the field methods and include the BUGS code!

Line 55: These statements about sampling methods for different amphibian stages are overly general. Although larval amphibians are typically sampled with nets or aquatic traps, I am unfamiliar with spotlight surveys for them. Even among terrestrial salamanders, sampling methods for adults can vary widely. The point about divergent methods for different life stages nonetheless is true for most amphibians with aquatic larvae and terrestrial adults.

We have modified the sentence to improve clarity

Lines 71–78: This paragraph would be better placed in the Materials and Methods.

We have moved most of the paragraph to the methods section.

Line 79: A paragraph here about the objectives of the research or the research questions addressed would bring the broader context of the Introduction to a logical conclusion regarding the impetus for this specific study.

We have more clearly stated our objectives in the penultimate paragraph of the introduction

Lines 87–88: See previous statement. It seems odd to reference the Introduction when referring to methods.

We have included more detailed field methods

Line 106–112: I recommend specifying the units for the parameters so that readers can better interpret the constraints imposed by the priors on lines 108–110.

added

Line 112: I recommend indicating that L ∞ refers to the asymptotic size of adults across all presented von Bertalanffy growth models if that is the case.

added

Lines 115–117: How is measurement error in this formulation separated from individual heterogeneity in growth, and how could these two sources of variation in L t be separated without marked individuals? It seems to me individual heterogeneity and measurement error would both be included in the variance parameter.

Good catch! this should’ve been in the model for adult growth; we have moved it accordingly 

Line 134: Is equivalent to L t on line 126? If so, I recommend changing L t on line 126 to to make this explicit.

corrected

Line 150: It is probably worth citing the R2WinBUGS package here as well.

done

Line 159: Delete the “s” in “reductions.”

Corrected

Line 167: Replace “shows” with “showed” to keep the results in past tense.

Corrected

Line 192: I recommend replacing “parameterizations” with “life stages” to place the emphasis on the biology of flatwoods salamanders.

Done

Lines 192–193: The credible limits in Figure 4 appear to go from 20 mm to 50 mm. Why is the value in the text so much narrower? It almost seems like the dashed lines on the figure are the posterior predictive interval, rather than the credible interval. The same comment applies on line 195 for asymptotic length.

Good catch! Corrected.

Line 201: Replace “reflect” with “is.”

Corrected

Lines 207–210: How was longevity determined, given the plateau in size at 7 years, variability in individual size, and measurement error? A growth model with this shape seems ill-suited for estimating maximum lifespan; the mark-recapture data could, however, be used in other ways to try to estimate longevity (for an example, see Fellers et al. 2013).

Yes there is a high degree of uncertainty in the age of larger individuals, and we agree this is not the best way to estimate longevity (longevity was more of a byproduct from the growth analysis). We have another manuscript in prep that is more focused on estimating survival rates / longevity explicitly as you suggest, however we feel this very rough estimate derived here from the growth increments is worth including, if only to act as a straw man for subsequent analysis. We have shortened this section so that is not as prominent in the manuscript.

Lines 254–264: Better description of how the potential lifespan of flatwoods salamanders was estimated is necessary, otherwise this paragraph is largely speculative.

Added details for clarity

Lines 265–283: This paragraph can be deleted. It is entirely based on unpublished data and a personal communication, and not on the model or results presented in this study.

removed

---

## [Decision Letter · Decision Letter 1]

15 Jul 2020

PONE-D-20-12247R1

Reconciling larval and adult sampling methods to model growth across life-stages

PLOS ONE

Dear Dr. Brooks,

Thank you for submitting your manuscript to PLOS ONE. After careful consideration, we feel that most of the reviewers' comments have now been addressed, but one reviewer has requested a number of further clarifications and improvements. Therefore, we invite you to submit a revised version of the manuscript that addresses the points raised during the review process.

Just one of the previous reviewers was available and willing to provide input on the revised version of your manuscript. This reviewer was not completely in agreement with some of the revisions provided and asked for several further points to be addressed and clarified.

We look forward to receiving your revised manuscript.

Kind regards,

William J. Etges

Academic Editor

PLOS ONE

Reviewers' comments:

Reviewer's Responses to Questions

**Comments to the Author**

1. If the authors have adequately addressed your comments raised in a previous round of review and you feel that this manuscript is now acceptable for publication, you may indicate that here to bypass the “Comments to the Author” section, enter your conflict of interest statement in the “Confidential to Editor” section, and submit your "Accept" recommendation.

Reviewer #2: (No Response)

2. Is the manuscript technically sound, and do the data support the conclusions?

Reviewer #2: Yes

3. Has the statistical analysis been performed appropriately and rigorously? 

Reviewer #2: Yes

4. Have the authors made all data underlying the findings in their manuscript fully available?

Reviewer #2: Yes

5. Is the manuscript presented in an intelligible fashion and written in standard English?

Reviewer #2: Yes

6. Review Comments to the Author

Reviewer #2: General Comments:

NOTE: I reviewed a previous version of this manuscript, and I found the revised version substantively improved. In reviewing the responses to reviewers, however, I noticed that my general comments were not included. I apologize if this was an oversight of mine when submitting the review. Below, I include general comments that remain relevant, but I have removed any that were addressed in revisions (many were requests for more details about field methods that were addressed in the revisions). All specific comments from the previous version of the manuscript were adequately addressed.

This manuscript addresses two important issues related to modeling population growth of organisms with complex life-histories: 1) the inability to accurately age individuals of many species, and 2) quantifying changes in growth across life stages of such organisms. The manuscript presents a model that provides estimates of parameters from larval growth through metamorphosis to adult growth. Such a model is very appealing, because it allows estimation of growth parameters across life-history stages to encompass an organism’s entire lifespan. Being able to model growth across life stages can be particularly important for parameterizing population models to develop conservation strategies. Thus, the topic of this manuscript is both interesting and important.

Although the authors have improved the Materials and Methods by providing many of the details regarding the drift fence sampling, some additional information is necessary. In particular, for studies that rely on recaptures of individuals, it is important to indicate how individuals identified or marked. Also, because of the centrality of size measurements for a growth study, it is important to indicate how larvae and adults were measured for readers to assess whether it is appropriate to share an error term (especially if it is specifically measurement error, and not variation around the mean estimated size—see specific comment below) across life stages in the model.

In addition to these general comments, I reference specific comments by line number below.

Specific Comments:

Lines 78: I recommend replacing the comma with “and” because the structure of the last element of the list (“variability in the timing and size at metamorphosis”) results in only two elements.

Line 89: Delete the period after “cm” and begin “funnel” with a lower case “f” if the preceding fragment is the size of the funnel traps.

Line 96: How was SVL measured? Was the same measurement method used for larvae (following paragraph)? I recommend including these details as they are relevant when trying to model growth across life-stage transitions.

Lines 122, 136, & 156: The equations on these lines appear to be missing a negative sign in front of the growth coefficient. It correctly appears on line 145.

Lines 138–139: tm is technically not the age at metamorphosis, but the time since the individual was theoretically length 0, correct? Is that incorporated into the prior, and is interpretation and subsequent use (e.g., in the adult growth equations) of tm as age at metamorphosis adjusted for the difference between hatching date and theoretical age at length 0? Reporting the posterior distribution of t0 also would help to explain how influential this technicality is: the closer t0 is to 0, the more closely tm will correspond to the age at metamorphosis.

Line 161: It is unclear to me how σ_L^2 separates measurement error from individual variation in growth. It seems to me that this error term would account for any variation between the model-estimated size, Lt, and the observed length, Lobs. Yes, this would include measurement error, but would it not also include individual heterogeneity? Even if individuals were measured perfectly, one would expect that Lobs ≠ Lt, as is the case for residual error when nearly any model is model fitted to data.

Lines 166–169: If I am reading this correctly, the number of retained posterior samples was 400,000 x 3 / 100 = 12,000. What was the minimum effective sample size across sampled parameters?

7. PLOS authors have the option to publish the peer review history of their article (what does this mean?). If published, this will include your full peer review and any attached files.

Reviewer #2: No

---

## [Author Response · Author response to Decision Letter 1]

28 Jul 2020

We would like to start by thanking the reviewers for their comments. I think they have greatly improved the manuscript. We apologize for missing these general comments in the first round of revisions.

This manuscript addresses two important issues related to modeling population growth of organisms with complex life-histories: 1) the inability to accurately age individuals of many species, and 2) quantifying changes in growth across life stages of such organisms. The manuscript presents a model that provides estimates of parameters from larval growth through metamorphosis to adult growth. Such a model is very appealing, because it allows estimation of growth parameters across life-history stages to encompass an organism’s entire lifespan. Being able to model growth across life stages can be particularly important for parameterizing population models to develop conservation strategies. Thus, the topic of this manuscript is both interesting and important.

Thanks, we think so too!

Although the authors have improved the Materials and Methods by providing many of the details regarding the drift fence sampling, some additional information is necessary. In particular, for studies that rely on recaptures of individuals, it is important to indicate how individuals identified or marked. 

We have added the marking method

Also, because of the centrality of size measurements for a growth study, it is important to indicate how larvae and adults were measured for readers to assess whether it is appropriate to share an error term across life stages in the model.

We have added the measuring method

Specific Comments:

Lines 78: I recommend replacing the comma with “and” because the structure of the last element of the list (“variability in the timing and size at metamorphosis”) results in only two elements.

corrected

Line 89: Delete the period after “cm” and begin “funnel” with a lower case “f” if the preceding fragment is the size of the funnel traps.

good catch! corrected

Line 96: How was SVL measured? Was the same measurement method used for larvae (following paragraph)? I recommend including these details as they are relevant when trying to model growth across life-stage transitions.

added measuring methods

Lines 122, 136, & 156: The equations on these lines appear to be missing a negative sign in front of the growth coefficient. It correctly appears on line 145.

corrected

Lines 138–139: tm is technically not the age at metamorphosis, but the time since the individual was theoretically length 0, correct? Is that incorporated into the prior, and is interpretation and subsequent use (e.g., in the adult growth equations) of tm as age at metamorphosis adjusted for the difference between hatching date and theoretical age at length 0? Reporting the posterior distribution of t0 also would help to explain how influential this technicality is: the closer t0 is to 0, the more closely tm will correspond to the age at metamorphosis.

The posterior distribution for t0 was very narrow, and approximately centered on zero, so doesn’t unduly influence interpretation. Despite this, we retained t0 in the adult growth equations for completeness. We have added a sentence to the results clarifying this. 

Line 161: It is unclear to me how σ_L^2 separates measurement error from individual variation in growth. It seems to me that this error term would account for any variation between the model-estimated size, Lt, and the observed length, Lobs. Yes, this would include measurement error, but would it not also include individual heterogeneity? Even if individuals were measured perfectly, one would expect that Lobs ≠ Lt, as is the case for residual error when nearly any model is model fitted to data.

you are correct; this is a relict from when we actually thought we could partition these two sources of uncertainty. I have clarified this where relevant. 

Lines 166–169: If I am reading this correctly, the number of retained posterior samples was 400,000 x 3 / 100 = 12,000. What was the minimum effective sample size across sampled parameters?

The lowest ESS across parameters was approximately 600. I have added a sentence to the results stating this.

---

## [Editor Report · Decision Letter 2]

3 Aug 2020

Reconciling larval and adult sampling methods to model growth across life-stages

PONE-D-20-12247R2

Dear Dr. Brooks,

We’re pleased to inform you that your manuscript has been judged scientifically suitable for publication and will be formally accepted for publication once it meets all outstanding technical requirements.

Kind regards,

William J. Etges

Academic Editor

PLOS ONE
---

## [Editor Report · Acceptance letter]

4 Aug 2020

PONE-D-20-12247R2 

Reconciling larval and adult sampling methods to model growth across life-stages 

Dear Dr. Brooks:

I'm pleased to inform you that your manuscript has been deemed suitable for publication in PLOS ONE. Congratulations! Your manuscript is now with our production department. 

Kind regards, 

on behalf of

Dr. William J. Etges 

Academic Editor

PLOS ONE